# Simultaneous Determination of Tetracyclines and Fluoroquinolones in Poultry Eggs by UPLC Integrated with Dual-Channel-Fluorescence Detection Method

**DOI:** 10.3390/molecules26185684

**Published:** 2021-09-19

**Authors:** Yawen Guo, Zhaoyuan He, Jinyuan Chen, Lan Chen, Kaizhou Xie, Tao Zhang, Genxi Zhang, Guojun Dai

**Affiliations:** 1College of Animal Science and Technology, Yangzhou University, Yangzhou 225009, China; dx120200135@yzu.edu.cn (Y.G.); mz120191027@yzu.edu.cn (Z.H.); mz120191044@yzu.edu.cn (J.C.); zhangt@yzu.edu.cn (T.Z.); gxzhang@yzu.edu.cn (G.Z.); daigj@yzu.edu.cn (G.D.); 2Joint International Research Laboratory of Agriculture & Agri-Product Safety, Yangzhou University, Yangzhou 225009, China; dz120200026@yzu.edu.cn; 3College of Veterinary Medicine, Yangzhou University, Yangzhou 225009, China

**Keywords:** tetracyclines, fluoroquinolones, dual-channel, UPLC-FLD, poultry eggs

## Abstract

An innovative, rapid and stable method for simultaneous determination of three tetracycline (oxytetracycline, tetracycline and doxycycline) and two fluoroquinolone (ciprofloxacin and enrofloxacin) residues in poultry eggs by ultra-high performance liquid chromatography–fluorescence detection (UPLC-FLD) was established and optimized. The samples were homogenized and extracted with acetonitrile/ultrapure water (90:10, *v/v*) and then purified by solid-phase extraction (SPE). LC separation was achieved on an ACQUITY UPLC BEH C18 column (1.7 µm, 2.1 mm × 100 mm), and the mobile phase was composed of acetonitrile and a 0.1 mol/L malonic acid solution containing 50 mmol/L magnesium chloride (the pH was adjusted to 5.5 with ammonia). When the five target drugs were spiked at the limit of quantification, 0.5 times the maximum residue limit (MRL), 1.0 MRL and 2.0 MRL, the recoveries were above 83.5% and the precision ranged from 1.99% to 6.24%. These figures of merit complied with the parameter validation regulations of the EU and U.S. FDA. The limits of detection and quantifications of the targets were 0.1–13.4 µg/kg and 0.3–40.1 µg/kg, respectively. The proposed method was easily extended to quantitative analyses of target drug residues in 85 egg samples, thus demonstrating its reliability and applicability.

## 1. Introduction

At present, high-density, intensive production determined by explosive increases in human consumption has led to the extensive use of antibiotics during breeding to treat numerous illnesses that may affect the growth, health and production of edible animals. However, to improve animal welfare, production of free-range eggs has increased worldwide. For instance, approximately 15% of laying hens in the EU are kept free range [1]. Free-range egg production and the entire breeding process increase the potential for contact with the external environment, which could increase disease risk for laying hens and thus reliance on protection from antibiotics.

Antibiotic residues in animal-derived foods have always been of widespread concern and easily cause cross-border trade disputes. However, there are environmentally friendly ways to handle such residues, such as biogasification or conversion into bonsai materials and other crafts. However, most of the waste byproducts, such as viscera, bones, feathers and eggshells, are directly discarded or processed into protein feed, while the excreta discharged are untreated or used as organic fertilizers [2]. Eventually, antimicrobials and their metabolites and isomers bioaccumulate up the food chain to humans. Tetracycline (TC) drugs, especially oxytetracycline (OTC), can chelate the calcium needed for growth, resulting in calcium deficiencies and delayed bone growth. The structures of TC drugs contain active groups that bind proteins strongly; therefore, such drugs can cause liver damage and gastrointestinal discomfort. The accumulation of fluoroquinolones in vivo may cause abnormal nerve stimulation because they have a configuration similar to that of γ-aminobutyric acid, which is a neurotransmission inhibitor. The fluorine contained in fluoroquinolones can interfere with the activity of bone alkaline phosphatase and then cause tendon inflammation and cartilage dysplasia. Assuredly, these abovementioned toxic effects are not easily triggered by the low levels of residues in animal products. Both tetracyclines and fluoroquinolones could cause allergic reactions; the former is more prone to cause skin diseases, asthma and even shock, while the latter may cause skin discomfort and phototoxicity. In addition to the treatment and prevention of diseases, antibiotics have the distinct economic advantage of promoting growth; that is, speeding up the farm-to-fork process. Nonetheless, the EU [3] and the USA [4] banned the use of antibiotic growth promoters in 2006 and 2017, respectively, and banning or promoting alternatives to this type of antibiotic is also on the agenda in China. The transfer of antimicrobial resistance in zoonotic bacteria has been confirmed, which means that the therapeutic effects of originally reliable drugs have been limited [5]. The TC resistance levels in *Escherichia coli* isolates from broilers are approximately 40% in China, and the levels of resistance to ciprofloxacin (CIP) and enrofloxacin (ENR) are 62% and 71%, respectively [6]. Resistance to fluoroquinolones develops more slowly than that to TCs, because the former is chromosomally mediated rather than plasmid mediated [7]. The problems of antimicrobial resistance caused by long-term use, overuse and abuse of drugs and unquantifiable discharges of residues pose an unpredictable threat to public health, driving up health care costs and generating persistent environmental risks.

Therefore, it is necessary to monitor and trace residues in animal-derived foods. The means of monitoring and tracing depend on the establishment of reliable determination methods, and the development and optimization of such methods should consider the edible tissues and the corresponding maximum residue limits (MRLs). China stipulates that the MRLs of OTC and TC in poultry eggs are both 400 μg/kg [8], while the EU’s regulation is 200 μg/kg [9]. Both China and the EU have banned the use of doxycycline (DOX), CIP and ENR during the laying period. This study uses the MRLs of the three abovementioned drugs in poultry muscle, namely, 100 μg/kg specified by the EU, as the data basis. Importantly, banning use does not prevent noncompliant abuse. In addition, although residual ENR is defined as the sum of the residual prototype drug and its metabolite CIP, considering the complex metabolism in poultry and the deposition in eggs, the two were studied independently. Currently reported LC methods for simultaneous determination of TCs and fluoroquinolones include FLD and diode array detection (DAD). Compared with DAD, FLD exhibits lower background noise and higher sensitivity, and both excitation and emission radiation have bandwidth selectors. There is only one UPLC-FLD method reported to date [10], but the targets were not the same as those of this study, and the sample matrixes and sample pretreatment method were also different. The aim of this study was to establish a dual-channel method for the simultaneous determination of OTC, TC, DOX, CIP and ENR in poultry eggs (chicken eggs and duck eggs) by UPLC-FLD. Compared with albumen, yolk has a longer development period, and the albumen is laid down over 2–3 h after the yolk matures [11]. Because the five target drugs have different hydrophobicities, abilities to bind plasma proteins and metabolic transformation capabilities in laying hens, they are generally preferentially deposited in the yolk. Hence, blended whole egg, albumen and yolk should be divided into distinct matrix samples when detecting drug residues in poultry eggs.

## 2. Materials and Methods

### 2.1. Chemicals and Reagents

OTC (CAS No. 79-57-2, purity ≥ 97.0%) and TC (CAS No. 60-54-8, purity ≥ 98.0%) standards were purchased from Sigma-Aldrich LLC (St. Louis, MO, USA). A DOX (CAS No. 564-25, purity ≥ 98.0%) standard was acquired from Merck Drugs & Biotechnology Co., Inc. (Fairfield, OH, USA). CIP (CAS No. 85721-33-1, purity ≥ 98.0%) and ENR (CAS No. 93106-60-6, purity ≥ 98.0%) standards were obtained from Macklin Inc. (Shanghai, China).

LC-grade methanol and acetonitrile were provided by Tedia Company Inc. (Fairfield, OH, USA). Sodium hydroxide, citric acid, magnesium chloride, malonic acid and other reagents were of analytical grade and were supplied by Sinopharm Chemical Reagent Co. Ltd. (Shanghai, China). Sodium dodecyl sulfate was obtained from Solarbio Life Science Co., Ltd. (Beijing, China). Ultrapure water (18.2 MΩ* cm, 25 °C), which met the standard of ‘water for analytical laboratory use-specification and test methods’ [12], was prepared with a Nanopure water purifier produced by Thermo Fisher Scientific Inc. (Waltham, MA, USA). All solutions used in the LC system were filtered through a 0.45 μm pore size polytetrafluoroethylene microporous membrane filter and then degassed.

### 2.2. Stock and Working Solutions

Stock solutions of OTC, TC and DOX were prepared at 1 mg/mL with LC-grade methanol, divided, sealed and stored in an ultralow temperature freezer at −70 °C. The three TC standards tend to produce static electricity when contacting a dispensing spoon and are thus not easily removed from the spoon. OTC, TC and DOX stock solutions (1 mg/mL) were dissolved in 0.03 mol/L sodium hydroxide and brought to 10 mL with LC-grade methanol in a brown volumetric flask. TC stock solutions were prepared fresh monthly, and fluoroquinolone stock solutions were prepared fresh every 6 months.

The working solutions were pretreated by gradually diluting the stock solutions with LC-grade acetonitrile to the corresponding concentration required for subsequent testing and blank sample spiking. Fluoroquinolone working solutions were stored at 4 °C and prepared fresh monthly. In contrast to the long-term stability of the fluoroquinolone working solutions, the TC working solutions were unstable and easily degraded under light and high-temperature conditions. Thus, the latter were stored at 4 °C and prepared fresh daily while avoiding exposure to light and high temperatures.

### 2.3. UPLC-FLD Analysis

A Waters ACQUITY UPLC system (Waters Corp., Milford, MA, USA) integrated with a column manager, binary solvent manager, sample manager and other components was combined with a Waters fluorescence detector (Waters Corp., Milford, MA, USA). Each sample was loaded onto an ACQUITY UPLC BEH C18 (1.7 µm, 2.1 mm × 100 mm) column to achieve effective separation. The front end of the column was connected with a VanGuard BEH C18 column with the same inner diameter and particle size as the first column, and an ACQUITY in-line filter (Waters Corp., Milford, MA, USA) with a pore size of 0.2 μm. The column was heated to 35 °C. The injection volume was 10 μL. To avoid exceeding the pressure limit of the system, the flow rate was set to 0.2 mL/min. The mobile phase consisted of LC-grade acetonitrile (A) and a 0.1 mol/L malonic acid solution containing 50 mmol/L magnesium chloride (B), and the pH was adjusted to 5.5 with ammonia (S210-B, Mettler Toledo LLC, Columbus, OH, USA). The gradient elution program was as follows: 0, 86% B; 2.5 min, 86% B; 7.0 min, 50% B; 8.0 min, 86% B; and 9.0 min, 86% B. The detector had an acquisition speed of 80 Hz and was set to 3D scanning. For channel A, the excitation wavelength (Ex) for OTC, TC and DOX was 416 nm, and the emission wavelength (Em) was 518 nm. For channel B, the Ex for CIP and ENR was 274 nm, and the Em was 428 nm. The detection wavelengths were set according to the scanning result of a multifunction microplate reader (EnSpire, PerkinElmer Inc., Singapore). Relevant parameters and procedures were set by Empower 3 software (Waters Corp., Milford, MA, USA) with excellent data integrity and traceability.

### 2.4. Sample Collection and Pretreatment

The entire breeding process, which minimizes the stresses caused by environmental changes and personnel actions, was carried out in strict accordance with the recommendations of the Guidelines for the Care and Use of Experimental Animals in Jiangsu Province, and was authorized by the Ethics Review Board of Yangzhou University. Forty Haiyang yellow chickens (Jiangsu Jinghai Poultry Industry Group Co., Ltd., Nantong, Jiangsu, China) aged 30 weeks and forty Gaoyou ducks (Jiangsu Gaoyou Duck Corp., Yangzhou, Jiangsu, China) aged 30 weeks were randomly selected. Before collecting eggs, the chickens were fed complete formula feed without any added drugs for 14 days. A collection period of 20 days was selected in September to include the peak laying period of the ducks. The intake of complete formula feed without any added drugs and clean water was ad libitum, and eggs were collected from 17:30 to 18:00 every day. After collection, the blended whole egg, albumen and yolk were separated and homogenized (PT-3100, Polytron Technologies Inc., Luzern, Switzerland), packed into a 50 mL centrifuge tube, labeled and stored at −34 °C until analysis.

Homogenized blank samples (2.0 ± 0.02 g) were precisely weighed with an analytical balance (AX205, Mettler Toledo LLC, Columbus, OH, USA) into 50 mL centrifuge tubes, and 10 mL of acetonitrile/ultrapure water (90:10, *v/v*) was added. The mixed solution was vortexed on a Vortex-Genie 2 mixer (SI-0246, Scientific Industries Inc., Bohemia, NY, USA) for 2 min. After ultrasonic extraction for 10 min using an ultrasonic cleaner (Elmasonic P300H, Elma Electronic Ltd., Munich, Germany), the mixed solution was centrifuged for 10 min in a desktop high-speed refrigerated centrifuge (5810R, Eppendorf Corp., Hamburg, Germany), the rotating speed was installed at 9000× *g*, and the temperature was set to 4 °C. The supernatant was transferred to a new centrifuge tube. The muddy residues at the bottom were re-extracted as described above, and the two supernatants were combined.

The collected supernatants were run through Waters Oasis PRiME HLB SPE columns (60 mg/3 mL, Waters Corp., Milford, MA, USA). Three milliliters of methanol, 3 mL of ultrapure water and 3 mL of sodium dodecyl sulfonate buffer were added sequentially to activate and equilibrate the column. After that, the collected supernatant was injected, 3 mL of initial mobile phase (A:B = 14:86, *v/v*) was loaded for rinsing, and 2 mL of methanol was added to elute the target drugs. The subatmospheric pressure generated by a vacuum pump (JTCQ-24D, Jtone Electronic Co. Ltd., Hangzhou, Jiangsu, China) dominated the droplet velocity during SPE and prevented possible blockage of the SPE column by the sample extract.

The eluate was evaporated to near dryness by a steam dryer (N-Evap 112, Organomation Corp., Berlin, MD, USA) supplied by a nitrogen generator (Genius 1024, Peak Scientific Instruments Ltd., Inchinnan, Scotland, UK), and the tray temperature was set at 40 °C. The key to nitrogen purging is that the eluate in the tube should not be blown dry. Before injection into the UPLC-FLD system, 2 mL of mobile phase was added to the residues obtained in the prior step, which were resuspended by vortexing for 3 min and filtered through a 0.22 μm Millipore Millex-HV sterile filter (Merck KGaA Co. Inc., Darmstadt, Hesse, Germany).

### 2.5. Method Validation

#### 2.5.1. Linearity

The standard working solutions of five target drugs were diluted to a series of solutions with different concentrations by pipetting an appropriate amount of matrix extract from a blank sample so that the final spiked concentrations corresponding to the blank sample (blended whole chicken egg, chicken albumen, chicken yolk, blended whole duck egg, duck albumen and duck yolk) were at the limit of quantification (LOQ) and 100.0, 200.0, 400.0, 600.0, 800.0 and 1000.0 µg/kg for OTC; at the LOQ and 100.0, 200.0, 400.0, 600.0, 800.0 and 1000.0 µg/kg for TC; at the LOQ and 50.0, 100.0, 150.0, 200.0, 250.0 and 300.0 µg/kg for DOX; at the LOQ and 50.0, 100.0, 150.0, 200.0, 250.0 and 300.0 µg/kg for CIP; and at the LOQ and 50.0, 100.0, 150.0, 200.0, 250.0 and 300.0 µg/kg for ENR. The linear standard curves for the five target drugs in different sample matrixes were drawn using the spiked concentration as the independent variable (x) and the detected peak area as the dependent variable (y).

#### 2.5.2. Recovery

Next, (2.0 ± 0.02) g of homogenized egg sample was accurately weighed, and appropriate amounts of OTC, TC, DOX, CIP and ENR standard working solutions were added so that the concentrations of the target drugs in each blank sample were at the LOQ, 0.5 MRL, 1.0 MRL and 2.0 MRL, and each concentration was prepared with 6 parallel replicates. Afterwards, the sample was extracted, cleaned up, concentrated and reconstituted according to the sample pretreatment process described in Section 2.4. The peak area of the pretreated filtrate detected by UPLC-FLD was substituted into the linear regression equation described in the previous paragraph to calculate the detected concentration. The ratio of the detected concentration to the actual spiked concentration was the recovery.

#### 2.5.3. Precision

The precision was evaluated from the relative standard deviation (RSD), which is the ratio of the SD to the mean concentration, and was divided into intraday precision (or within-run precision) and interday precision (or between-run precision). At different times on the same day, the same instrument and the same standard curve were used to analyze the samples spiked with four concentrations (LOQ, 0.5 MRL, 1.0 MRL and 2.0 MRL), and each spiked concentration was set up in 6 parallel replicates to calculate the intraday RSD. The interday RSD was determined from four spiked concentrations on different days of the week with a newly drawn standard curve used each day to investigate the standard curve, instrument performance and environmental changes and other small random fluctuations.

#### 2.5.4. Sensitivity

The limit of detection (LOD) and LOQ were calculated from the signal-to-noise ratio (S/N). When S/N ≥ 3 and 10, the actual spiked concentrations corresponded to the LOD and LOQ, respectively.

## 3. Results and Discussion

### 3.1. Optimization of the Sample Pretreatment

The difficulty of sample pretreatment is determined by the high dynamic range, complex chemical properties and heterogeneity of compounds within animal-derived food matrices and the susceptibility of the LC-FLD system. The selection of extractant should take into account the target drug selectivity and deproteinization, and the extractant should have good tissue permeability to release drugs that are associated with tissue. Organic solvents, such as acetonitrile and ethyl acetate, are often applied to precipitate protein during extraction of protein-rich poultry eggs. Mobile phase A was LC-grade acetonitrile, so to avoid having to switch to other solvents that may have caused detection interferences, acetonitrile/ultrapure water was selected to extract the target drugs. This study compared the extraction efficiencies of acetonitrile and ultrapure water in different volume ratios of 80:20, 85:15, 90:10 and 95:5. The recovery increased as the acetonitrile/ultrapure water volume ratio increased, but there was little difference between 90:10 and 95:5. To reduce solvent consumption and organic matter volatilization during nitrogen blowing, acetonitrile/ultrapure water with a volume ratio of 90:10 was chosen as the extractant. Acidified acetonitrile is often used to extract drugs from egg samples. In the method established by Frenich et al. for the determination of five drugs (TCs and quinolones) in eggs by tandem mass spectrometry (MS/MS), the extractant consisted of acetonitrile, citric acid (pH 4) and Na_2_EDTA solutions [13]. Sodium succinate buffer was used as an extractant in the study described by Heller et al. [14]. In the report by Piatkowska et al., the extractant used in egg pretreatment was 0.1% formic acid in acetonitrile/water containing EDTA [15]. The extractant used in the present study (acetonitrile/ultrapure water) achieved good extraction efficiency, which may be attributed to the sufficient mixing of the extractant and the sample during liquid–liquid extraction (LLE) and the reduced loss of the target drugs during SPE.

SPE, as the cleanup step after LLE, has the advantages of no emulsification, low solvent consumption and effective separation of target drugs from interferences. Different types of SPE columns are commercially available. The choice of SPE column and the solvent used in each step determine the extraction efficiency. As relatively new adsorbents for SPE columns, high-molecular-weight polymers, such as those in the Oasis MCX column, Oasis MAX column, Oasis hydrophilic–lipophilic balance (HLB) column and Oasis PRiME HLB column, have greatly enhanced extraction efficiencies. Oasis MCX is a mixed strong cation exchange adsorbent with high selectivity for alkaline analytes. Oasis Max is a mixed strong anion exchange adsorbent with high selectivity for acidic analytes. The Oasis HLB adsorbent is suitable for alkaline, acidic and neutral analytes. Heller et al. [14] and Piatkowska et al. [15] used SPE columns with Oasis HLB adsorbent. The Oasis PRiME HLB adsorbent not only has the water wettability and reversed-phase retention characteristics of Oasis HLB, but also simplifies the extraction procedure by omitting the activation and equilibrium steps. The recoveries of fluoroquinolones from the Oasis MCX column and CIP from the Oasis MAX column were low, and the removal of interferences by the Oasis PRiME HLB column was better than that by the Oasis HLB column. A C18 column, typically applied for cleanup, was also compared, but the recoveries of TCs were insufficient. Consequently, Oasis PRiME HLB columns were utilized for sample cleanup. Although the Oasis PRiME HLB adsorbent does not need to be activated or equilibrated, these two steps were carried out in this study for recovery comparison. The capacity of the SPE column determined from the volume of the supernatant after extraction was 60 mg/3 mL. Unlike manual cleanup, in which controlling the oscillation frequency and reducing operational variations are difficult, the main advantage of SPE is that the procedure has high repeatability. The sorbent, which has a strong specific adsorption capacity, is so tightly packed into an SPE column that the presence of suspended matter in a sample can easily cause blockage; thus, extracts must be relatively clean prior to SPE. Moreover, the flow rate is not easy to control, and the subatmospheric pressure generated via a vacuum pump may cause the filler to collapse. However, these drawbacks have not affected the popularization of SPE applications, and its extraction efficiency is also reflected in the excellent removal of biological matrix components, especially proteins and phospholipids, which lead to obvious matrix effects [16].

Filter membranes composed of three typical organic materials were also compared: polyvinylidene fluoride (PVDF), polytetrafluoroethylene, and polypropylene. A Millipore Millex-HV sterile filter made of PVDF was selected. The pore size of the membrane was 0.22 μm, which met the requirements of the UPLC system. This filter membrane has stable quality and enables smooth filtration and strong binding to particles, gases and microorganisms, and the thread of the edge ensures that the filter will not slip.

### 3.2. Optimization of UPLC Separation and FLD

The keto and enol groups in TCs constitute a conjugated double bond system, which determines their fluorescence properties [17]. The rigid planar structure (two juxtaposed six-membered rings) of fluoroquinolones with their large π-conjugated system composed of benzoheterocycles and chromophores, such as carbonyl groups, carboxyl groups and heteroatoms, endow these compounds with high intrinsic fluorescence [18]. Therefore, the target drugs can be detected directly by FLD and do not require derivatization. Ex and Em are FLD parameters that directly affect the sensitivity and selectivity of detection. The optimal Ex and Em of different target drugs are not the same, so compromises are necessary when setting these parameters. In the studies of Schneider et al. using an HPLC-FLD method to simultaneously detect fluoroquinolones and TCs in catfish muscle [19] and chicken muscle [20], the Ex and Em were set to 375 nm and 535 nm, respectively, for TCs and 275 nm and 425 nm, respectively, for fluoroquinolones. In Castillo-García et al.’s study on the simultaneous determination of fluoroquinolones and TCs in animal muscles (pig muscle and chicken muscle) by dispersive SPE-UPLC-FLD, the Ex and Em were set to 390 nm and 512 nm, respectively, for TCs, and to 255 nm and 360 nm, respectively, for fluoroquinolones [10]. In this study, the optimal Ex and Em of each target drug were determined by scanning the detection wavelengths of the microplate reader, and then the sensitivity for each target drug was comprehensively considered. The optimal Ex (416 nm) and Em (518 nm) of TC were set as the detection wavelengths for all TCs. The optimal Ex (274 nm) and Em (428 nm) for CIP were set as the detection wavelengths for fluoroquinolones.

After the detection wavelengths of the fluorescence detector are optimized, the optimal chromatographic separation conditions need to be determined. We evaluated common mobile phases in different LC-FLD methods: the organic phase is generally methanol or acetonitrile, and the aqueous phase generally contains formic acid [13,14], heptafluorobutyric acid [15] or ammonium acetate [10]. However, the peaks of the three TCs coeluted and could not be completely separated. Referring to the research of Schneider et al. [19,20], we attempted to use a malonic acid solution as the mobile phase. When the concentration was adjusted to 0.1 mol/L, all the target peaks were completely separated, but the TC peak shapes were poor. Then, 50 mmol/L magnesium chloride was added to improve the peak shapes and reduce the elution time. Schneider et al. considered that the addition of magnesium ions could enhance the fluorescence intensity of TCs [19,20]. Calcium ions were added for the HPLC-DAD method established by Moudgil et al. [21]. When methanol was used as the mobile phase, the baseline noise and the solvent peaks in channel B were obvious and interfered with the TC peaks. In addition, because the elution capacity of an acetonitrile–water system is stronger than that of a methanol–water system, the use of acetonitrile as the mobile phase results in sharper target peaks and higher sensitivity. A low pH of the mobile phase may cause dissociation of the silanol groups on the stationary phase of the chromatographic column, so the pH not only affects the fluorescence intensity of targets but also affects the retention of targets in the column and the peak separation. TCs and fluoroquinolones are both amphoteric: TCs are easily degraded in low-pH solutions, the fluorescence intensity of fluoroquinolones in low-pH solutions is weak, and the peaks of both compound classes are prone to tailing. This study compared different pH values (4.5, 5.0, 5.5 and 6.0) for mobile phase B and found that 5.5 was the most appropriate. This pH was much lower than that (6.5) reported by Schneider et al. [19,20] and that (6.9) reported by Castillo-García et al. [10], likely because of the different chromatographic columns used.

An Agilent ZORBAX Eclipse XDB-phenyl column (3.5 μm, 3.0 mm × 150 mm) was used by Schneider et al. [19,20], a Thermo Scientific Syncronis C18 column (1.7 μm, 2.1 mm × 100 mm) was applied by Castillo-García et al. [10], and a Waters ACQUITY UPLC BEH C18 (1.7 µm, 2.1 mm × 100 mm) was adopted in the present experiment. In comparison with the column used by Schneider et al., the column that we used had a smaller particle size, which meant greater flow resistance, so our flow rate of 0.2 mL/min was lower than the 0.5 mL/min set by Schneider et al. [19,20].

Taking chicken egg samples as an example, the chromatograms obtained with the optimized UPLC separation and FL detection conditions are shown in Figure 1, Figure 2, Figure 3, Figure 4, Figure 5 and Figure 6. Each matrix (blended whole chicken egg, albumen and yolk) has chromatograms corresponding to channel A and channel B. Although the chromatograms from the two channels for the same matrix are visualized separately for a more intuitive comparison between the standards and spiked blanks, the data from the two channels were acquired simultaneously. Comparison of the chromatograms from the same channel for the blanks and blanks spiked with standards shows that the shape of each target peak is sharp, the peaks are well separated, and the baseline is stable. However, comparing the chromatograms from channel A for each matrix reveals that the TC and DOX peaks coelute with the solvent peak.

### 3.3. Validation

OTC, TC, DOX, CIP and ENR were spiked in blended whole chicken egg, chicken albumen, chicken yolk, blended whole duck egg, duck albumen and duck yolk at concentrations of the LOQ-1000.0 µg/kg, LOQ-1000.0 µg/kg, LOQ-300.0 µg/kg, LOQ-300.0 µg/kg and LOQ-300.0 µg/kg, respectively. The peak area (y-axis) of each target had a good linear relationship with the spiked concentration (x-axis), and all the determination coefficients (R^2^) were greater than or equal to 0.9992. The linear ranges, linear regression equations and R^2^ are shown in Table 1.

The five target drugs were spiked into blended whole chicken egg, chicken albumen, chicken yolk, blended whole duck egg, duck albumen and duck yolk at the LOQ, 0.5 MRL, 1.0 MRL and 2.0 MRL.The recovery and precision data for the five target drugs are shown in Table 2, Table 3, Table 4, Table 5, Table 6 and Table 7. As seen in Table 2, Table 3, Table 4, Table 5, Table 6 and Table 7, the recoveries of the five target drugs were 83.50–95.58%, the intraday RSDs were 1.99–4.91%, and the interday RSDs were 2.10–6.24%. The method validation protocol, parameters and data are in line with the EU 2002/675/EC resolutions [22] and U.S. FDA guidelines [23], in which acceptable recoveries for tests of multidrug residues are 7–120%.

The LODs and LOQs of the five targets determined in blank poultry eggs using the newly established detection method are presented in Table 8. The LODs of the three TCs and two fluoroquinolones were 5.2–13.4 µg/kg and 0.1–0.5 µg/kg, respectively, and the LOQs were 17.4–40.1 µg/kg and 0.3–1.5 µg/kg, respectively.

### 3.4. Applications

A total of 85 commercial poultry eggs (60 chicken eggs and 25 duck eggs) from different vendors and various breeds of chickens and ducks were purchased from various local markets. The eggs were divided into blended whole eggs, albumen and yolk, then homogenized, processed with the optimized pretreatment method and analyzed by the described UPLC-FLD method. Only two duck egg samples (50.1 µg/kg in blended whole duck egg and 54.0 µg/kg in duck yolk) were found to contain TC residue, but the residual concentrations did not exceed the EU regulatory limit of 200 μg/kg [9]. The dual-channel-UPLC-FLD method was easily extended to quantification in real egg samples, which sufficiently demonstrates its practicability and reliability.

### 3.5. Comparison with Previously Reported Methods

To date, the reported methods for the simultaneous detection of TCs and fluoroquinolones consist of mostly LC tandem mass spectrometry, and the analyzed matrixes include environmental samples, such as soil [24] and sewage sludge [25], eggs [13,14,15] and other edible animal tissues [26,27]. In general, the targets of LC-MS/MS methods are not limited to TCs and fluoroquinolones. The expense associated with MS techniques limits their application and popularization, and the development of LC techniques can provide supplemental and alternative methods to simultaneous detection methods. FLD and DAD are used in combination with LC [21,28,29], and different detectors have different detection principles. Table 9 presents a comprehensive comparison with previously reported LC methods. Additionally, compared with LC-DAD detection methods, this simultaneous detection method has a shorter analysis time. The studies of Schneider et al. on catfish muscle [19] and chicken muscle [20] did not include SPE purification, and the analysis times were much longer than that of the present study. Another distinction is the difference between LC separation systems, and the advantages of UPLC over HPLC are related to the optimization of monolithic systems and faster separation efficiency. For example, the matrix, sample pretreatment method, mobile phase and detection wavelength were different from those in the research of Castillo-García et al. [10]. Furthermore, the targets were not exactly the same: the targets in the work of Castillo-García et al. included three TCs (OTC, TC and chlortetracycline) and three acidic quinolones (oxolinic acid, nalidixic acid and flumequine) [10]. On the other hand, compared with HPLC systems marketed earlier, UPLC improves the separation efficiency, quantitation ability and peak response sensitivity by reducing the particle sizes of the column filling material and enhancing the system’s resistance to high pressure while reducing analysis time and solvent consumption. Overall, the established and optimized dual-channel-UPLC-FLD method provides novel techniques for the simultaneous determination of TC and fluoroquinolone residues in poultry eggs.

## 4. Conclusions

An LLE-SPE-dual-channel-UPLC-FLD method was established and optimized for the simultaneous determination of three tetracycline residues (oxytetracycline, tetracycline and doxycycline) and two fluoroquinolone residues (ciprofloxacin and enrofloxacin) in poultry eggs. The proposed method is accurate and precise, has high recoveries and complies with parameter validation regulations of the EU and U.S. FDA. The developed method was easily extended to quantitative analyses of the target drugs at concentrations below their corresponding MRLs in real poultry egg samples, which demonstrates the reliability and practicality of the method.

## Figures and Tables

**Figure 1 molecules-26-05684-f001:**
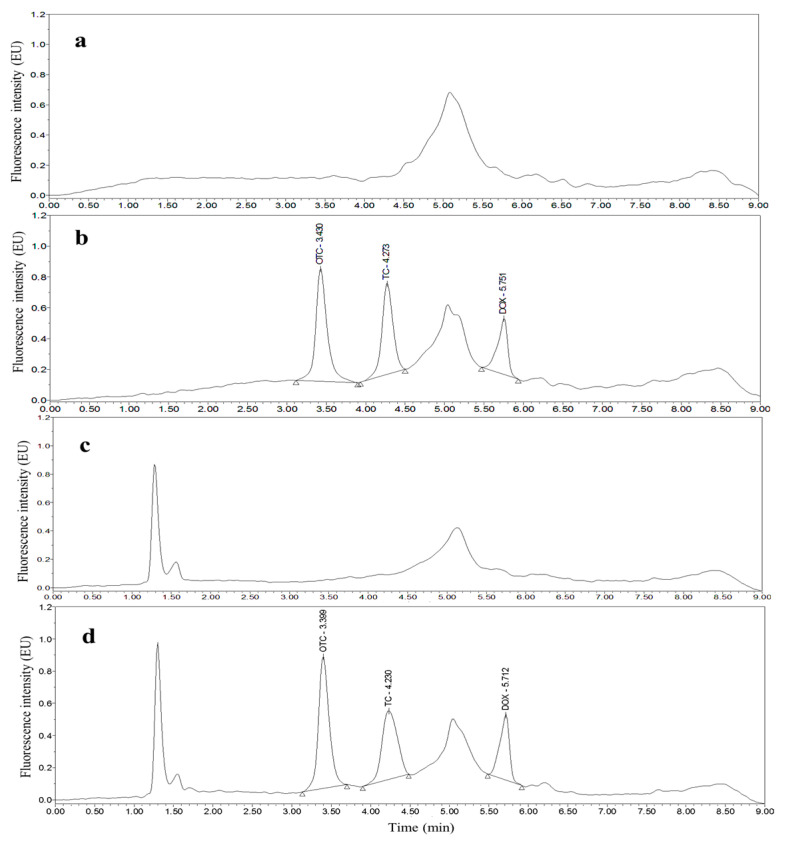
Chromatograms of the mobile phase (**a**), 200 µg/kg OTC, TC and DOX standards (**b**), blank blended whole chicken egg (**c**) and blank blended whole chicken egg spiked with 200 µg/kg OTC, TC and DOX standards (**d**) from channel A.

**Figure 2 molecules-26-05684-f002:**
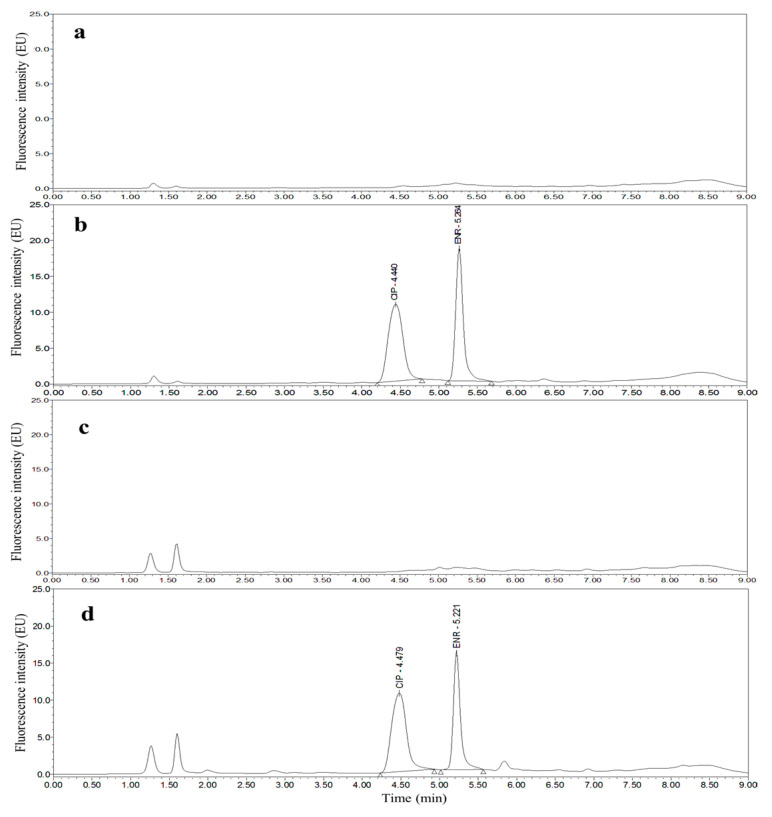
Chromatograms of the mobile phase (**a**), 50 µg/kg CIP and 25 µg/kg ENR standards (**b**), blank blended whole chicken egg (**c**) and blank blended whole chicken egg spiked with 50 µg/kg CIP and 25 µg/kg ENR standards (**d**) from channel B.

**Figure 3 molecules-26-05684-f003:**
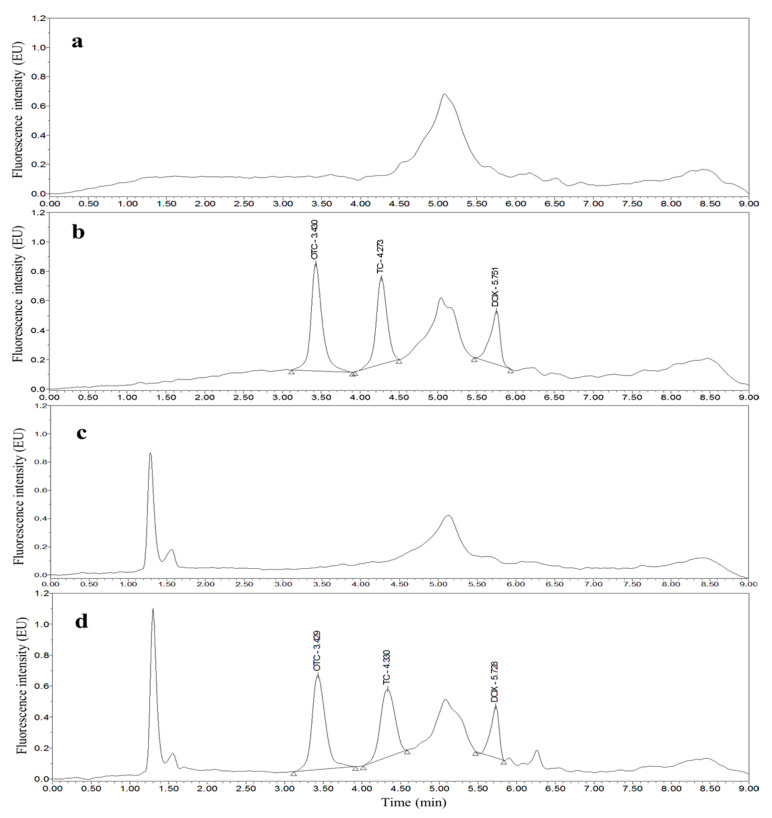
Chromatograms of the mobile phase (**a**), 200 µg/kg OTC, TC and DOX standards (**b**), blank chicken albumen (**c**) and blank chicken albumen spiked with 200 µg/kg OTC, TC and DOX standards (**d**) from channel A.

**Figure 4 molecules-26-05684-f004:**
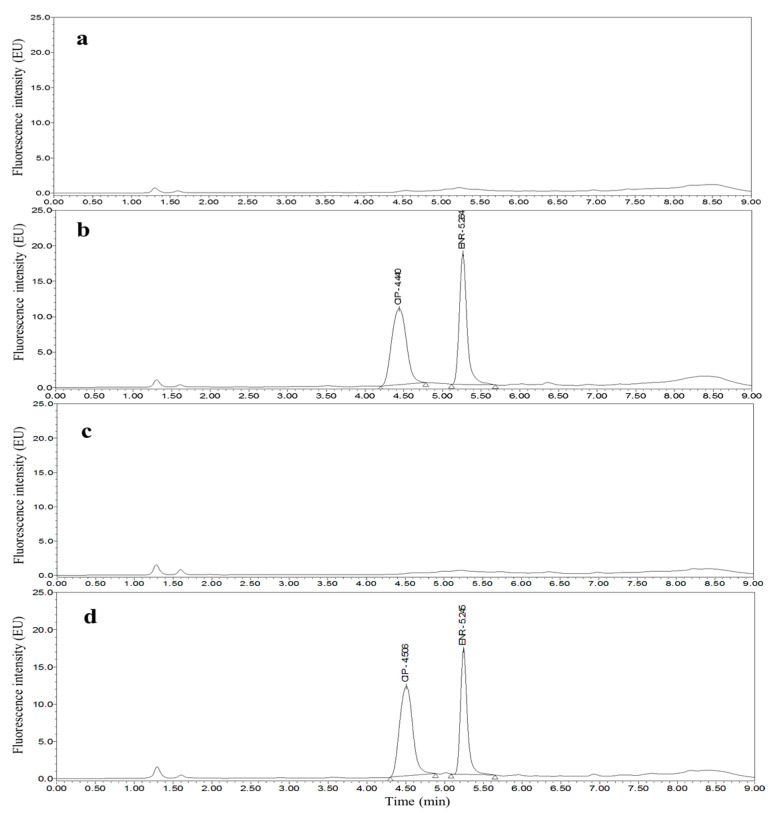
Chromatograms of the mobile phase (**a**), 50 µg/kg CIP and 25 µg/kg ENR standards (**b**), blank chicken albumen (**c**) and blank chicken albumen spiked with 50 µg/kg CIP and 25 µg/kg ENR standards (**d**) from channel B.

**Figure 5 molecules-26-05684-f005:**
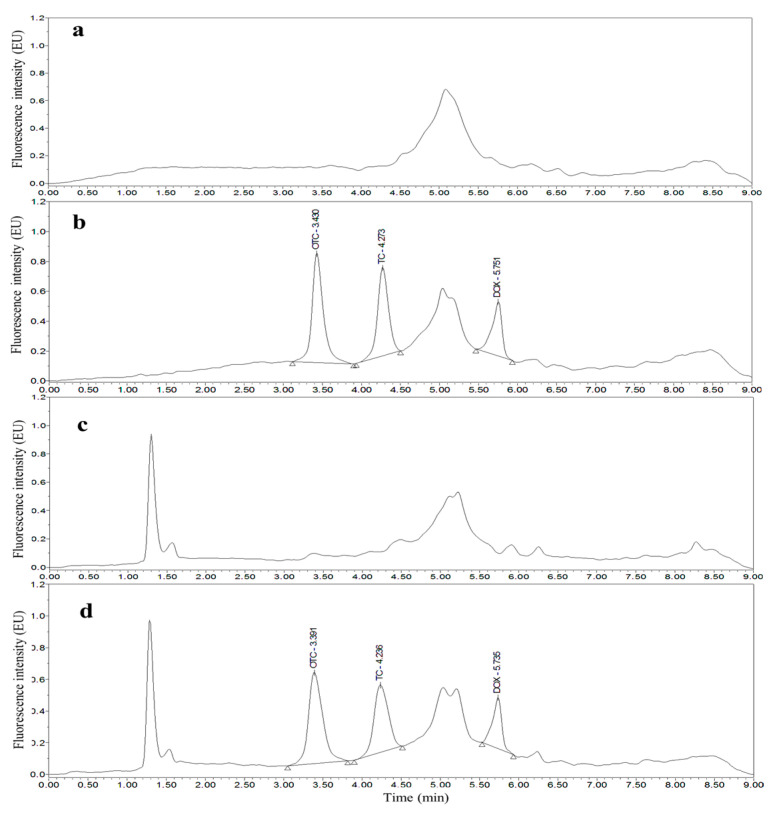
Chromatograms of the mobile phase (**a**), 200 µg/kg OTC, TC and DOX standards (**b**), blank chicken yolk (**c**) and blank chicken yolk spiked with 200 µg/kg OTC, TC and DOX standards (**d**) from channel A.

**Figure 6 molecules-26-05684-f006:**
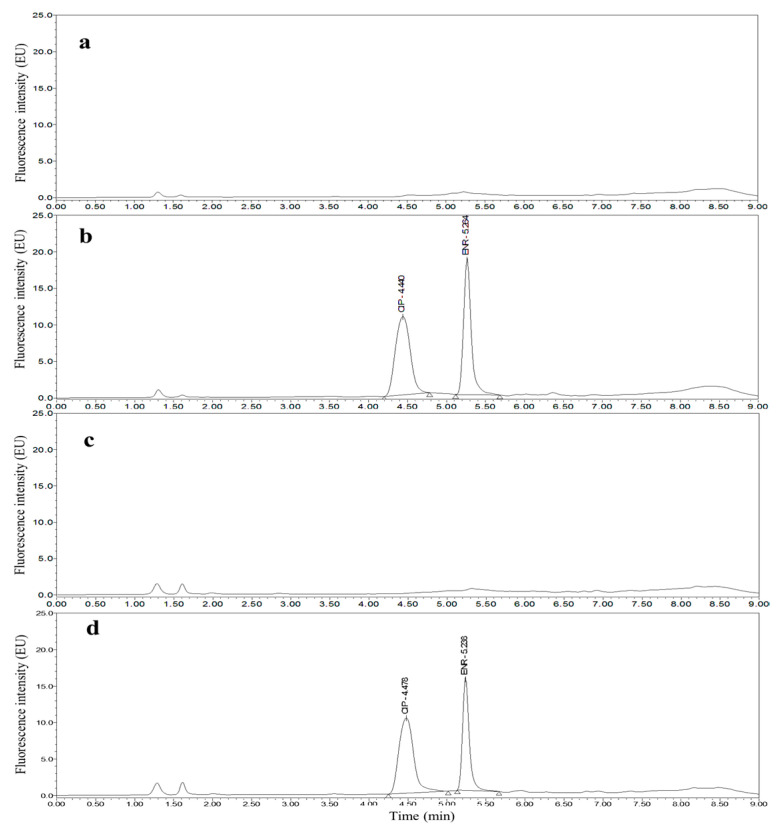
Chromatograms of the mobile phase (**a**), 50 µg/kg CIP and 25 µg/kg ENR standards (**b**), blank chicken yolk (**c**) and blank chicken yolk spiked with 50 µg/kg CIP and 25 µg/kg ENR standards (**d**) from channel B.

**Table 1 molecules-26-05684-t001:** Linearity ranges, linear regression equations and determination coefficients for determinations of OTC, TC, DOX, CIP and ENR in poultry eggs.

Matrix	Analyte	Linearity Range (µg/kg)	Linear Regression Equation	Determination Coefficient (R^2^)
Blended whole chicken egg	OTC	19.2–1000.0	y = 425.51x − 2034.3	0.9998
	TC	31.2–1000.0	y = 360.26x + 1118.4	0.9998
	DOX	35.3–300.0	y = 248.98x + 7889.8	0.9993
	CIP	0.6–300.0	y = 35325x − 9347	0.9998
	ENR	0.4–300.0	y = 39161x + 282018	0.9998
Chicken albumen	OTC	17.4–1000.0	y = 418.55x − 1723.3	0.9997
	TC	28.8–1000.0	y = 396.38x − 3708.4	0.9994
	DOX	31.9–300.0	y = 281.71x + 3259.9	0.9992
	CIP	0.7–300.0	y = 33285x + 10819	0.9996
	ENR	0.3–300.0	y = 37248x + 398456	0.9996
Chicken yolk	OTC	25.6–1000.0	y = 449.03x − 6847.2	0.9994
	TC	36.0–1000.0	y = 363.79x − 4823.8	0.9993
	DOX	40.1–300.0	y = 286.29x − 71.871	0.9998
	CIP	1.2–300.0	y = 35999x + 30938	0.9996
	ENR	0.3–300.0	y = 43943x + 166512	0.9994
Blended whole duck egg	OTC	20.1–1000.0	y = 476.11x − 5044.9	0.9999
	TC	32.1–1000.0	y = 359.6x + 1599.6	0.9998
	DOX	34.7–300.0	y = 302.9x − 2627.5	0.9995
	CIP	1.1–300.0	y = 36124x − 20065	0.9997
	ENR	0.5–300.0	y = 44043x + 124503	0.9997
Duck albumen	OTC	19.2–1000.0	y = 444.68x − 968.85	0.9996
	TC	27.3–1000.0	y = 394.95x − 7178.2	0.9994
	DOX	32.5–300.0	y = 296.54x − 815.06	0.9994
	CIP	0.7–300.0	y = 35712x + 126377	0.9998
	ENR	0.4–300.0	y = 39981x + 118026	0.9995
Duck yolk	OTC	21.6–1000.0	y = 413.89x + 538.36	0.9997
	TC	38.3–1000.0	y = 333.34x − 3420.4	0.9998
	DOX	37.2–300.0	y = 281.82x + 29.831	0.9995
	CIP	1.5–300.0	y = 40759x + 79433	0.9998
	ENR	0.3–300.0	y = 39750x + 220046	0.9993

**Table 2 molecules-26-05684-t002:** Recovery and precision for analyses of blank blended whole chicken egg spiked with OTC, TC, DOX, CIP and ENR (n = 6).

Analyte	Spiked Level (µg/kg)	Recovery (%)	RSD (%)	Intraday RSD (%)	Interday RSD (%)
OTC	19.2	86.93 ± 1.39	1.60	2.59	2.78
	200	86.90 ± 2.56	2.95	3.71	3.83
	400 ^α^	89.43 ± 1.63	1.82	2.03	2.32
	800	91.15 ± 3.21	3.52	4.52	4.72
TC	31.2	86.58 ± 2.88	3.32	4.14	4.26
	200	87.40 ± 1.70	1.95	3.85	3.57
	400 ^α^	89.28 ± 2.48	2.78	2.85	3.89
	800	90.20 ± 1.38	1.53	2.33	2.58
DOX	35.3	86.33 ± 2.25	2.60	3.61	3.13
	50	85.10 ± 3.01	3.54	3.10	3.94
	100 ^α^	85.15 ± 1.87	2.19	3.17	3.70
	200	84.65 ± 3.72	4.40	3.98	4.59
CIP	0.6	87.65 ± 2.29	2.60	3.49	3.90
	50	89.88 ± 2.23	2.48	3.62	3.41
	100 ^α^	92.35 ± 2.16	2.34	2.46	3.99
	200	92.88 ± 2.41	2.60	2.05	3.27
ENR	0.4	94.05 ± 1.88	1.99	3.07	4.17
	50	88.63 ± 2.49	2.80	2.93	3.61
	100 ^α^	95.58 ± 3.68	3.85	3.41	3.90
	200	92.63 ± 2.57	2.78	3.15	3.74

Note: the superscript α indicates maximum residue limits, which are the same below.

**Table 3 molecules-26-05684-t003:** Recovery and precision for analyses of blank chicken albumen spiked with OTC, TC, DOX, CIP and ENR (n = 6).

Analyte	Spiked Level (µg/kg)	Recovery (%)	RSD (%)	Intraday RSD (%)	Interday RSD (%)
OTC	17.4	87.28 ± 1.76	2.01	2.57	3.96
	200	86.33 ± 1.07	1.24	2.04	2.86
	400 ^α^	89.70 ± 2.60	2.90	3.15	3.09
	800	89.03 ± 3.30	3.71	4.28	4.32
TC	28.8	85.60 ± 3.15	3.69	3.82	3.95
	200	85.20 ± 1.53	1.79	2.24	3.40
	400 ^α^	90.05 ± 3.40	3.77	2.68	4.84
	800	89.28 ± 2.42	2.70	2.67	3.64
DOX	31.9	86.20 ± 2.15	2.50	3.41	3.57
	50	86.40 ± 1.54	1.79	2.23	2.37
	100 ^α^	87.53 ± 2.00	2.29	2.99	3.07
	200	89.58 ± 1.67	1.86	2.13	2.44
CIP	0.7	88.45 ± 2.65	2.99	2.59	3.69
	50	88.13 ± 1.72	1.96	2.32	2.66
	100 ^α^	90.15 ± 1.22	1.35	1.99	2.10
	200	90.78 ± 1.96	2.16	2.69	2.54
ENR	0.3	86.15 ± 3.52	4.09	4.67	6.24
	50	91.48 ± 1.78	1.94	2.36	4.18
	100 ^α^	89.58 ± 1.51	1.69	2.65	2.81
	200	91.45 ± 3.36	3.67	4.31	4.94

**Table 4 molecules-26-05684-t004:** Recovery and precision for analyses of blank chicken yolk spiked with OTC, TC, DOX, CIP and ENR (n = 6).

Analyte	Spiked Level (µg/kg)	Recovery (%)	RSD (%)	Intraday RSD (%)	Interday RSD (%)
OTC	25.6	85.30 ± 2.17	2.54	2.95	4.10
	200	86.53 ± 1.35	1.56	3.62	3.91
	400 ^α^	87.93 ± 2.40	2.73	2.94	4.92
	800	88.85 ± 2.96	3.33	3.35	4.12
TC	36.0	86.58 ± 2.88	3.32	3.55	3.97
	200	86.70 ± 1.16	1.34	3.29	3.11
	400 ^α^	88.05 ± 2.00	2.27	2.56	3.99
	800	89.15 ± 1.07	1.20	2.77	3.63
DOX	40.1	86.08 ± 1.83	2.12	2.35	3.97
	50	86.03 ± 2.70	3.15	3.63	4.71
	100 ^α^	85.15 ± 1.87	2.20	3.17	3.77
	200	85.13 ± 2.79	3.28	3.89	4.28
CIP	1.2	87.38 ± 1.89	2.17	2.38	3.89
	50	89.63 ± 1.99	2.22	3.03	3.82
	100 ^α^	91.85 ± 2.45	2.66	3.19	3.66
	200	92.88 ± 2.41	2.60	3.05	3.27
ENR	0.3	90.73 ± 1.70	1.88	2.07	3.37
	50	90.38 ± 2.00	2.20	2.74	3.30
	100 ^α^	95.58 ± 3.68	3.85	4.11	5.60
	200	93.88 ± 2.66	2.83	3.31	3.45

**Table 5 molecules-26-05684-t005:** Recovery and precision for analyses of blank blended whole duck egg spiked with OTC, TC, DOX, CIP and ENR (n = 6).

Analyte	Spiked Level (µg/kg)	Recovery (%)	RSD (%)	Intraday RSD (%)	Interday RSD (%)
OTC	20.1	85.33 ± 2.87	3.36	3.50	4.55
	200	85.78 ± 1.60	1.86	2.66	3.30
	400 ^α^	88.38 ± 1.88	2.13	2.91	3.11
	800	89.40 ± 2.50	2.81	3.85	4.19
TC	32.1	87.30 ± 2.15	2.46	3.61	4.48
	200	87.58 ± 2.18	2.49	3.20	3.55
	400 ^α^	88.35 ± 2.50	2.83	3.38	3.40
	800	89.18 ± 1.72	1.93	2.57	3.14
DOX	34.7	84.33 ± 2.68	3.18	3.62	4.51
	50	83.50 ± 1.10	1.32	3.12	3.31
	100 ^α^	86.15 ± 3.33	3.87	3.22	4.28
	200	84.83 ± 1.26	1.49	3.58	3.46
CIP	1.1	85.53 ± 2.05	2.40	3.51	4.17
	50	90.13 ± 2.66	2.95	3.68	4.14
	100 ^α^	91.80 ± 1.94	2.11	2.91	3.39
	200	92.03 ± 2.96	3.21	3.60	4.45
ENR	0.5	87.98 ± 3.12	3.55	3.46	4.01
	50	89.13 ± 2.65	2.97	3.29	3.04
	100 ^α^	94.35 ± 2.80	2.97	3.93	4.30
	200	92.63 ± 2.57	2.78	2.11	3.24

**Table 6 molecules-26-05684-t006:** Recovery and precision for analyses of blank duck albumen spiked with OTC, TC, DOX, CIP and ENR (n = 6).

Analyte	Spiked Level (µg/kg)	Recovery (%)	RSD (%)	Intraday RSD (%)	Interday RSD (%)
OTC	19.2	87.03 ± 2.14	2.46	3.57	4.22
	200	87.58 ± 1.82	2.08	3.83	3.54
	400 ^α^	88.48 ± 2.11	2.39	2.80	3.12
	800	88.18 ± 2.20	2.49	3.03	4.96
TC	27.3	84.10 ± 3.50	4.17	4.91	5.55
	200	86.33 ± 2.91	3.37	4.25	4.68
	400 ^α^	88.90 ± 3.66	4.12	4.06	5.28
	800	89.23 ± 2.12	2.38	3.86	3.37
DOX	32.5	84.45 ± 2.18	2.58	3.51	4.78
	50	85.88 ± 1.52	1.77	2.35	3.17
	100 ^α^	86.68 ± 3.40	3.92	3.96	5.80
	200	88.88 ± 1.79	2.02	3.76	3.69
CIP	0.7	88.45 ± 2.65	2.99	3.55	4.11
	50	87.85 ± 1.22	1.39	3.32	3.16
	100 ^α^	88.90 ± 3.53	3.98	4.70	5.33
	200	89.08 ± 1.60	1.79	2.07	3.88
ENR	0.4	87.90 ± 3.26	3.70	3.95	4.42
	50	90.98 ± 1.71	1.88	3.20	3.23
	100 ^α^	89.85 ± 2.05	2.28	3.23	4.77
	200	91.23 ± 3.50	3.84	2.88	4.27

**Table 7 molecules-26-05684-t007:** Recovery and precision for analyses of blank duck yolk spiked with OTC, TC, DOX, CIP and ENR (n = 6).

Analyte	Spiked Level (µg/kg)	Recovery (%)	RSD (%)	Intraday RSD (%)	Interday RSD (%)
OTC	21.6	86.93 ± 1.39	1.60	3.54	3.59
	200	86.85 ± 2.56	2.95	3.71	4.83
	400 ^α^	89.43 ± 1.63	1.82	2.13	3.32
	800	91.15 ± 3.21	3.52	3.86	4.72
TC	38.3	86.58 ± 2.88	3.32	3.14	4.26
	200	87.40 ± 1.70	1.95	2.95	2.57
	400 ^α^	89.28 ± 2.48	2.78	2.87	3.26
	800	90.20 ± 1.38	1.53	3.58	3.33
DOX	37.2	86.33 ± 2.25	2.60	3.25	4.13
	50	85.10 ± 3.01	3.54	3.48	5.10
	100 ^α^	87.65 ± 1.60	1.84	2.37	3.19
	200	90.90 ± 2.40	2.64	3.66	3.23
CIP	1.5	88.65 ± 3.30	3.72	4.11	4.52
	50	89.88 ± 2.23	2.48	2.62	3.41
	100 ^α^	92.35 ± 2.16	2.34	3.46	3.99
	200	92.63 ± 2.03	2.19	3.09	4.53
ENR	0.3	89.73 ± 1.42	1.59	2.40	3.66
	50	90.48 ± 1.60	1.78	2.82	3.86
	100 ^α^	92.51 ± 3.68	3.98	3.11	4.26
	200	92.63 ± 2.57	2.78	3.17	3.24

**Table 8 molecules-26-05684-t008:** LODs and LOQs of OTC, TC, DOX, CIP and ENR in poultry eggs.

Analyte	Matrix	LODs (µg/kg)	LOQs (µg/kg)
OTC	Blended whole chicken egg	5.5	19.2
	Chicken albumen	5.2	17.4
	Chicken Yolk	7.7	25.6
	Blended whole duck egg	6.0	20.1
	Duck albumen	5.4	19.2
	Duck yolk	6.5	21.6
TC	Blended whole chicken egg	9.1	31.2
	Chicken albumen	9.5	28.8
	Chicken yolk	10.6	36.0
	Blended whole duck egg	9.3	32.1
	Duck albumen	8.9	27.3
	Duck yolk	11.8	38.3
DOX	Blended whole chicken egg	10.5	35.3
	Chicken albumen	9.7	31.9
	Chicken yolk	13.4	40.1
	Blended whole duck egg	10.4	34.7
	Duck albumen	9.6	32.5
	Duck yolk	10.6	37.2
CIP	Blended whole chicken egg	0.2	0.6
	Chicken albumen	0.2	0.7
	Chicken yolk	0.4	1.2
	Blended whole duck egg	0.3	1.1
	Duck albumen	0.2	0.7
	Duck yolk	0.5	1.5
ENR	Blended whole chicken egg	0.1	0.4
	Chicken albumen	0.1	0.3
	Chicken yolk	0.1	0.3
	Blended whole duck egg	0.1	0.5
	Duck albumen	0.1	0.4
	Duck yolk	0.1	0.3

**Table 9 molecules-26-05684-t009:** Comparison with previously reported LC methods.

Detection Method	Sample Pretreatment Method	Analyte	Matrix	Analysis Time (min)	LOD (µg/kg)	LOQ (µg/kg)	Recovery (%)
UPLC-FLD (this study)	LLE-SPE	Tetracyclines and fluoroquinolones	Poultry eggs	9	0.1–13.4	0.3–40.1	83.50–95.58
UPLC-FLD [10]	Dispersive SPE	Tetracyclines and acidic quinolones	Pork and chicken muscle	6	0.25–3.8	-	61.5–102.6
HPLC-FLD [19]	LLE	Tetracyclines and fluoroquinolones	Catfish muscle	>25	-	0.15–1.5	60–92
HPLC-FLD [20]	LLE	Tetracyclines and fluoroquinolones	Chicken muscle	>25	-	0.5–5	63–95
UPLC-DAD [28]	Matrix solid-phase dispersion	Tetracyclines, fluoroquinolones and sulfonamides	Pork	14	0.5–3.0	-	74.5–102.7
HPLC-DAD [29]	Matrix solid-phase dispersion	Tetracyclines and fluoroquinolones	Porcine tissues	12	2–10	7–34	80.6–99.2
HPLC-DAD [21]	LLE-SPE	Tetracyclines, fluoroquinolones, sulfonamides and chloramphenicols	Milk	40	17.2–24.9	51.5–68.1	83.3–111.8

## Data Availability

All available data are contained within the article.

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
