# Peer review of "Simultaneous Determination of Tetracyclines and Fluoroquinolones in Poultry Eggs by UPLC Integrated with Dual-Channel-Fluorescence Detection Method"

_molecules, 2021, doi:10.3390/molecules26185684_

Round 1

Reviewer 1 Report

File attached

Author Response

Thank you for your encouragement. Your comments make us feel that our efforts expended during experimentation and writing are worthwhile. We are also very grateful for your suggestions, which are professional and directed to key points.

Reviewer 2 Report

The presentation is kind of messy. For example, we don’t use the term ‘method verification’ instead we use ‘method validation’. Some of the figure legends are written wrong (e.g. writing ENR as DOX) etc...

Introduction can be improved by describing the previous methods and the necessity of the current method. 

No mention of the ethical review board approval for performing the study. 

Section 2.5. needs to be divided into subsections to describe each component of method validation separately. 

I would like to refer the authors to look at FDA's guideline for bioanalytical method validation for proper method validation protocol. No description of carryover, ruggedness, robustness, and stability study under method validation. 

The authors did a good amount of work. However, there is plenty of room to clean up and improve their manuscript.

The authors might consider submitting to a lesser impact journal.

Author Response

We thank you for your work and your valuable comments and solutions. These comments provided the most direct and effective suggestions for improving this manuscript. 

Reviewer 3 Report

The research described in the paper is adequately planned and properly performed. Although I do not find much novelty (it is just an application of the existing method to a new matrix) and some parts are (what is the point of separate methods for whites and yolks?), I could recommend accepting the paper after introducing the corrections described below.

Some statements in the introduction may be misleading:

  1. The described toxic effects of antibiotics relate to therapeutic doses and do not apply to the residue levels. The main problem with antibiotic residues is AMR and this should be emphasized. Possibly, a passage on the allergies can be added.
  2. Some countries do study the usage of antibiotics in different groups of farm animals (please refer to ESVAC).
  3. There are direct proofs of transfer of AMR in zoonotic bacteria (please refer eg. to 10.3201/eid1601.090729).

There are mistakes or at least doubtful approach also in Materials and methods:

  1. Why EDTA was not used for sample extraction? The efficiency of the extraction could then depend a lot on the contents of metal cations in the sample.
  2. Oasis PRiME HLB is a reversed-phase SPE so applying the analytes in 90% MeCN would result in total loss of them. Some steps must be omitted in the description.
  3. Why was so high a spiking level chosen for fluoroquinolones? They are not allowed in laying hens and the method LOQ would allow for lower levels.

Please consider the following changes for Results:

  1. Reduce the number of Figures. Really, there is no need of presenting each matrix.
  2. Could you please explain the peak at RT ca 5.0 min occurring in the tetracyclines chromatogram? It is a bit odd for me. First, I do not expect the solvents to elute so late. Second, the FLD detection is rather selective. Was appropriate purity of solvents used?
  3.  

Author Response

Thank you so much for your valuable and professional comments and for taking the time to review this manuscript.

Round 2

Reviewer 2 Report

The authors improved the manuscript. It can be accepted in its current form. 

Reviewer 3 Report

The authors have responded to most of my remarks. Therefore, I can recommend the approval of the submission after some minor corrections. 

First, please, remove the superscript relating to the MRLS for DOX, CIP, and ENR. It gives the impression that these MRLs have been introduced for the tested matrix, which is false.

Second, please consider further reducing the number of figures. In my opinion, two would be sufficient.